# The Association between Oral Health Management and the Functional Independence Measure Scores at the Time of Admission of Inpatients to a Convalescent Hospital

**DOI:** 10.3390/geriatrics8050104

**Published:** 2023-10-18

**Authors:** Ryuzo Hara, Naoki Todayama, Tomohiro Tabata, Miki Kuwazawa, Tomoko Mukai, Yukiko Hatanaka, Shouji Hironaka, Nobuyuki Kawate, Junichi Furuya

**Affiliations:** 1Department of Oral Function Management, Showa University Graduate School of Dentistry, Ota-ku, Tokyo 145-8515, Japan; r.hara@dent.showa-u.ac.jp (R.H.); ed20-n201@grad.showa-u.ac.jp (N.T.); gd22-t014@dent.showa-u.ac.jp (T.T.); miki@dent.showa-u.ac.jp (M.K.); t.mukai@dent.showa-u.ac.jp (T.M.); y.hatanaka@dent.showa-u.ac.jp (Y.H.); 2Fujigaoka Hospital Hospitaly Dentistry, Yokohama-shi 227-8501, Japan; 3Department of Oral Hygiene, Showa University Graduate School of Dentistry, Ota-ku, Tokyo 145-8515, Japan; hironaka@dent.showa-u.ac.jp; 4Department of Rehabilitation Medicine, Showa University Graduate School of Medicine, Ota-ku, Tokyo 142-8555, Japan; kawate@med.showa-u.ac.jp

**Keywords:** convalescence hospital, oral health management, activities of daily living, nutrition, dental treatment

## Abstract

Many older patients admitted to convalescent hospitals present with impaired oral function, which is an important function of daily life. This study aimed to investigate the actual status of the oral healthcare needs of inpatients in a convalescent hospital and to clarify the relationship between the level of independence at admission and the oral function. The participants were 147 patients (94 males, 53 females, mean age: 74.6 ± 13.1 years) who received dental visits during their stay at a convalescent hospital. Information regarding general and oral health was extracted from medical records. Most patients were found to have low Functional Independence Measure motor scores, and approximately 70% had some form of oral intake, but approximately 80% had nutritional problems. The mean number of present and functional teeth were 16.6 and 20.8, respectively, and 65% of patients lost molar occlusal support. Multiple regression analysis showed significant positive correlations of high functional independence measure with age, eating status, nutritional status, and the number of functional teeth. This implied that oral health management is important for patients in a convalescent hospital and that enhancing oral health may be related to improved eating textures and better nutritional status.

## 1. Introduction

The older adult population in Japan is increasing every year, and the number of older adults requiring nursing care is also increasing accordingly. Stroke, joint diseases, dementia, fractures, and weakness account for 70% of the major causes of the need for nursing care [1]. Patients with these conditions receive rehabilitation at convalescent hospitals after being discharged from acute care hospitals and return to their residences when their daily functions improve.

Many patients in such convalescent hospitals have impaired activities of daily living (ADL) due to stroke or fracture and undergo intensive long-term rehabilitation to improve ADL during hospitalization. The Functional Independence Measure (FIM) is often used to make decisions regarding discharge after the recovery phase of rehabilitation. According to the latest data, the FIM motor items and total score cutoffs for the discharge of stroke patients are 70 and 82 points, respectively. The long-term rehabilitation schedule is determined based on the FIM score at admission [2,3,4,5,6].

It is often challenging to spend time on improving the oral health of patients with stroke in the acute phase owing to their intensive treatment schedules. Nonetheless, paralysis may prevent independent oral hygiene care, and the inability to intake food orally due to tube feeding may worsen oral health. Convalescent hospitals have longer hospital stays, up to 6 months, and are more likely to provide treatment other than the main disease, such as dental care, in between rehabilitation treatments. Improving oral health by providing dental treatment, especially prosthetic denture treatment to increase functional teeth, during hospitalization at convalescent hospitals is important, because once patients are discharged, it is generally difficult for them to visit the dentist independently [7].

Previous studies have demonstrated the importance of oral health management in patients with acute stroke through medical and dental cooperation [8,9]. However, denture treatment requires at least one to two months and is difficult to provide in the acute phase because of the short hospitalization period and various difficulties associated with treatment. Therefore, it is necessary to assess the dental treatment needs at the time of admission to the hospital [8]. In addition, analyses of the relationship between the oral and general health of patients admitted to convalescent hospitals have revealed that the deterioration in oral health is often associated with malnutrition, [9] eating habits, and oral health status [10]. In convalescent hospitals, many patients on parenteral nutrition often have poor oral health. Moreover, the negative cycle is likely to continue, as deteriorating oral health is likely to lead to the slow improvement in the ADL in rehabilitation. Studies on the associations between ADL and oral health in patients in convalescent hospitals have revealed that oral function, especially tongue pressure and the FIM score, is associated with oral intake in stroke patients [11]. In addition, there is evidence to suggest an association between oral health status and ADL in the convalescent hospital [12,13,14,15], although the specifics of this relationship are not clear. Oral health management is important in convalescent hospitals to support independence. However, oral health management needs and the association between oral and general health remain unknown [16].

Therefore, the purpose of this study was to clarify the oral health management needs within convalescent hospitals, as well as to investigate the association between FIM scores at admission and the oral health status.

## 2. Materials and Methods

### 2.1. Participants

A total of 154 patients with oral problems received dental visits during their hospitalization at a convalescent hospital between January 2019 and December 2020. Of these, 147 patients were included in this study, after excluding seven patients who met the exclusion criteria (namely, incomplete medical record data or diagnoses of brain diseases other than stroke). We extracted the information regarding the patient and oral cavity characteristics from the medical records and conducted a cross-sectional survey. Informed consent was obtained from all study participants before the study was initiated using the opt-out method. This study was approved by the Clinical Study Review Committee of Showa University Fujigaoka Hospital (approval number: F2020C158).

### 2.2. Outcomes

Information regarding the age, sex, main disease, independence (FIM motor, cognitive, and total scores), nutritional status (Controlling Nutritional Status [CONUT] variant score), eating status (Eating Status Scale [ESS] score), number of present teeth, number of functional teeth, Eichner classification, denture use, denture treatment needs, and physician requests were extracted from the medical records.

Upon admission to the convalescent rehabilitation ward, the dentist makes ward rounds to patients within one week to perform an oral examination and oral health status (Oral Health Assessment Tool [OHAT] score) scoring to determine the need for treatment.

General conditions were extracted from the medical records. The disease registered by the physician at the time of admission to the convalescent hospital was designated as the main disease. The FIM is an ADL evaluation instrument developed by Granger et al. in 1983 [17]. It consists of 18 motor and cognitive items, each rated on a 7-point scale (from 1 to 7)—13 motor items: I. Self-care, comprising (1) eating, (2) dressing, (3) wiping, (4) upper body changing, (5) lower body changing, and (6) toileting; II. Voiding control, comprising (7) urinary control and (8) bowel movement; III. Transfers, comprising (9) bed, chair, and wheelchair transfer, (10) toilet transfer, and (11) bathtub and shower transfer; and IV. Mobility, comprising (12) walking/wheelchair and (13) stairs, as well as five cognitive items: I. Communication, comprising (14) comprehension and (15) expression and II. Social cognition, comprising (16) social interaction, (17) problem-solving, and (18) memory.

The CONUT variant method was used to evaluate the nutritional status by replacing the total cholesterol with hemoglobin [18,19]. The serum albumin concentration (g/dL), total lymphocyte count (/µL), and hemoglobin concentration (mg/dL) were used to score the patients. The serum albumin level was scored on a 4-point scale—levels of ≥3.50 g/dL were scored as 0 (normal), 3.00–3.49 g/dL as 2 (mild), 2.50–2.99 g/dL as 4 (moderate), and <2.50 g/dL as 6 (severe). A total lymphocyte count of 1600/μL or more was scored as 0 points, 1200–1599/μL as 1 point, 800–1199/μL as 2 points, and less than 800/μL as 3 points. Hemoglobin concentration ≥13.0 mg/dL for males and ≥12.0 mg/dL for females were scored as 0 (normal), male 10.0–12.9 mg/dL and female 10.0–11.9 mg/dL as 1 (mild), 8.0–9.9 mg/dL as 2 (moderate), and <8.0 mg/dL as 3 (severe). These scores were added to obtain an overall total score (0–1 for normal, 2–4 for mild, 5–8 for moderate, and 9–12 for severe). 

The ESS was used to determine the nutritional intake status based on a 5-point scale: oral feeding (unmodified), oral feeding (modified), oral intake > tube feeding, oral intake < tube, and tube feeding only.

The number of functional teeth included natural teeth and the teeth in dentures but excluded the remaining roots and teeth with a mobility level of Grade III. The need for prosthetic treatment during hospitalization was classified into two groups: “presence” or “absence”. This classification represents the assessment of necessity by dentists and the patients’ demand for prosthetic treatment. Molar occlusal support was classified by Eichner classifications as follows: A1 to B3 were designated as having occlusal support of the molars, whereas those classified as B4 to C3 were designated as having no occlusal support of the molars [20]. Denture use was divided into two groups: denture wearing and not wearing due to ill-fitting or lack of dentures. The reasons for referral to dentists by medical doctors were categorized as follows: oral health examination for dysphagia, denture treatment, oral hygiene management, and others.

### 2.3. Statistical Analysis

A correlation analysis was used to analyze the correlation between the participants’ FIM scores and the respective factors. Multiple regression analysis was used to determine the association of level of independence, as the objective variable, with general and oral health, with the FIM score as the objective variable. SPSS ver. 27 (IBM Japan) was used for all statistical analyses, and the significance level was set at 5%.

## 3. Results

### 3.1. Participant Characteristics

Table 1 shows the mean ± SD or total number (%) results for each item. (Table 1) The mean age of the participants (94 males and 53 females) was 74.6 ± 13.1 years, and 83.7% of them were older than 65 years. Stroke was the main disease in the case of 73 participants (49.7%), followed by disuse syndrome in 30 cases (20.4%). The mean FIM motor item score was low (32.3 ± 18.9), while the mean cognitive item score was high (20.9 ± 10.1). The mean total score was 53.0 ± 26.0. Based on the ESS scores, food texture modification was not necessary in the case of 37 patients, whereas 72 patients required modified food due to a decline in masticatory and swallowing function. Most patients (74.2%) were taking some amount of food orally. The CONUT variant scores revealed that 85.0% of patients had malnutrition. The mean number of present and functional teeth were 16.6 ± 10.2 and 20.8 ± 9.6, respectively. Most of the patients had a dentition deficiency and required denture treatment due to ill-fitting dentures or lack of dentures. Occlusal support is important for oral intake of an unmodified diet; however, for the B4–C3 Eichner classification patients, 65.2% did not have dentures (Table 2). Furthermore, prosthetic treatment during hospitalization by the dentist was provided for only 36% of inpatients admitted to the convalescent hospital.

Regarding the reasons for referral to dentists from a medical doctor, 51.0% was for oral health examination, 36.1% was for denture treatment, and 10.2% was for oral hygiene management. (Figure 1).

### 3.2. Association between the Level of Independence and the Oral and General Condition

In the convalescent phase, rehabilitation is provided to improve the level of independence during hospitalization in order to make life after discharge as comfortable as possible. Therefore, with the level of independence as the outcome, we first analyzed the correlation between the level of independence and the general and oral health.

Correlation analysis showed that the FIM total score was positively correlated with age and ESS. The presence or absence of stroke showed a negative correlation. The FIM total score was significantly correlated with the presence of stroke and eating patterns but not with oral health status. The strongest correlation was with ESS. (Table 3). From these results, seven variables for multiple regression analysis were selected.

Multiple regression analysis was conducted with the FIM total score as the objective variable and age, sex, presence or absence of stroke, ESS score, CONUT variant score, number of functional teeth, and the presence or absence of denture treatment as the explanatory variables. There was a significant positive association between the FIM total score and age, ESS score, CONUT score, and the number of functional teeth (Table 4).

## 4. Discussion

The results of this study showed that many convalescent patients who were referred to a dentist had malnutrition despite the fact that many of them eat unmodified food orally. Additionally, many patients did not use dentures despite the lack of molar occlusal support, suggesting that oral problems might be the cause of the discrepancy between the eating status and malnutrition at the time of admission. In addition, many patients did not wish to have their dentures treated, even though they needed such treatment.

The reason for the poor nutritional status despite a good diet is presumably because patients suffering from stroke are often in poor general condition and may require feeding tubes. Additionally, since the participants of this study were patients with oral problems referred by medical departments, most of them had poor oral conditions. In other words, patients with low activities of daily living (ADL) may have easy access to nutrition regardless of their oral condition, while patients with high ADL may have relatively good eating patterns but may not be consuming enough food if they have oral problems. In fact, according to the table of molar occlusal support and denture availability (Table 2), many patients do not wear dentures despite the presence of molar defects. Although we did not conduct a survey on this issue, it is also unclear whether patients with dentures are using appropriate dentures. Therefore, it is possible that patients with poor eating status receive more nutritional management than those with good eating status. To further support this hypothesis, future studies should also consider the amount of food consumed.

The results of the correlation between the FIM total score and each of the factors showed no correlation with factors related to oral health. It is possible that because the patient in this case had oral problems and a dental request, the correlation may not have been influenced by a good or bad FIM score. The correlation results showed that age is positively correlated, which may be due to the lower average age of stroke patients and the fact that non-stroke patients, especially those suffering from fractures and spinal cord and spine disease, are older and more independent than stroke patients. From the results of this correlation, seven variables for multiple regression analysis were carefully selected. Seven was considered appropriate because of the 147 participants. As basic information, we extracted age, sex, stroke, and ESS with which we found a correlation, the CONUT variable as an indicator of nutrition, and the number of functional teeth and need for prosthetic treatment as indicators of oral health.

In addition, from the results of multiple regression analysis, the total FIM score was found to be associated with the number of functional teeth, eating status, and nutritional status, suggesting that improvement of each factor might contribute to the improvement in the FIM independently. Therefore, it is important to provide occlusal support, including dentures, as soon as possible for patients in convalescent hospitals. Although there were few requests for denture treatment in convalescent patients, the potential need for dentures was recognized. This highlights the importance of conducting oral health examinations for all convalescent patients and an active collaboration between medical and dental specialties to help patients who need dental intervention but do not request it on their own.

Previous studies have shown that patients with acute stroke benefit from oral health management. However, the average length of stay in acute care hospitals in Japan is 16.0 days, which is too short for prosthetic treatment such as dentures. Conversely, the maximum length of stay in convalescent hospitals is 180 days, which is more than enough time for denture treatment. This would mean that dentists could provide intensive oral health management for older people with declining ADL before they come back to their homes. 

Comparing previous studies [2,3,4,5,6], the primary disease in our study was the disease that impairs ADL and requires long-term rehabilitation and nursing care [1]. Therefore, their FIM motor and total scores were low. Some previous studies have shown that patients with malnutrition had poor oral health and that diet and nutrition were associated with oral health at the time of admission. Therefore, it is likely that oral health needs to be improved in order to improve the nutritional status [9,10,11]. Eating unmodified food would be important to maintain appetite, and loss of molar occlusal support might lead to a decline in appetite, resulting in malnutrition. In fact, many patients in our study did not have dentures even though they required dentures. Therefore, it is important for patients in convalescent hospitals to maintain and recover their occlusal support. However, most referrals from medical doctors were for oral health examination for dysphagia. It would be better if dentists routinely checked the oral health of hospitalized patients to determine the need for dental treatment. However, it must also be noted that this is often difficult because convalescent hospitals generally have only limited dental resources. As shown in previous studies, the oral health of patients admitted to convalescent hospitals is poor, and they often require some type of dental treatment as well as oral hygiene management [12,13,14,15]. The results of the multiple regression analysis clarified that the number of functional teeth is associated not only with the ability to eat regular food and maintain a good nutritional status but also with daily functioning. It is possible that the restoration of occlusion for inpatients with a reduced number of teeth at the time of admission may directly improve their ADL and indirectly help them improve their diet texture and nutrition by recovering proper masticatory and swallowing function.

It is, therefore clear that the need for denture treatment in the recovery period is high. This study only examined the relationship between FIM and general health and oral health status. Therefore, the direct relationship between nutritional status and eating status is unknown. A possible reason for this difference is that previous studies [10,11,12,13,14,21,22] evaluated nutritional status using the Mini-Nutritional Assessment-Short Form and oral intake using the Functional Oral Intake Scale (FOIS), but this study used the CONUT variant and the ESS instead. These correlations suggest that the MNA-SF score is associated with ADL, but the results could not be replicated because the FOIS was not evaluated in this study.

Some limitations of this study should be noted. The data used in this study other than prosthetic treatment were obtained only at the time of admission, and the degree of improvement is unknown because evaluation at discharge was not performed. In the future, it will be necessary to evaluate patients at the time of discharge to clarify improvements in oral health after dental treatment. In addition, the evaluation of the oral health was not performed on the day of admission but was referred by a medical doctor within approximately 10 days after admission, and it may be possible that rehabilitation progressed and the ADL improved by that time. Therefore, it is necessary to understand the schedule of the patients’ dental examinations and the number of rehabilitation procedures performed prior to these examinations. Furthermore, the results were obtained from a single recovery hospital, and bias can be expected as a result. A more comprehensive analysis can be conducted by collecting data from other facilities and evaluating patients at the time of admission and discharge. In addition, we did not specify the site of the stroke injury, and the relationship between the general condition and the oral cavity could also change depending on the fracture site. Therefore, it will be necessary in the future to classify patients according to the sites of stroke injury and fracture. There was also a difference between younger and older patients according to the age of the target patients, and restricting the age of the target patients may change the results. In addition, no differences were found in the oral health of the patients in this study, since only those patients were requested by their physicians, and all had oral problems. In the future, we would like to conduct a study on hospitalized patients to examine oral differences with and without intervention.

However, with these limitations clarified, our study has shown the association between the total FIM score, nutritional status, feeding status, and prosthetic treatment needs of patients admitted to convalescent hospitals. These findings may provide useful data to guide future endeavors to implement effective oral health management strategies, including dental treatment for patients admitted to convalescent hospitals.

## 5. Conclusions

Among the patients admitted to convalescent hospitals, a higher FIM was associated with a better diet texture, better nutritional status, and a higher number of functional teeth. This implied that oral health management is important for patients in a convalescent hospital and that enhancing oral health may be related to improved eating textures and better nutritional status. 

## Figures and Tables

**Figure 1 geriatrics-08-00104-f001:**
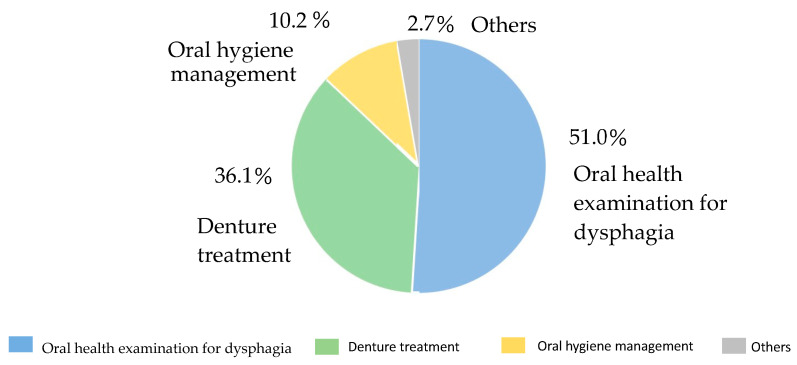
The reasons for referral to dentists from a medical doctor (%).

**Table 1 geriatrics-08-00104-t001:** Participant characteristics.

	Mean ± SD/N (%)
Age	74.6 ± 13.1
Main disease	
Stroke	73 (49.7)
Disuses syndrome	30 (20.4)
Fracture	15 (10.2)
Spinal cord disease	9 (6.1)
Others	20 (13.6)
FIM score	
Motor	32.3 ± 18.9
Cognitive	20.9 ± 10.1
Total	53.0 ± 26.0
ESS	
Oral feeding (unmodified)	37 (25.2)
Oral feeding (modified)	72 (49.0)
Tube feeding < oral intake	0 (0.0)
Tube feeding > oral intake	3 (2.0)
Tube feeding only	35 (23.8)
CONUT variant	
Severe	10 (6.8)
Moderate	49 (33.3)
Mild	66 (44.9)
Normal	22 (15.0)
Number of present teeth	16.6 ± 10.2
Number of functional teeth	20.8 ± 9.6
Molar occlusal support	
Present (EichnerA1-B3)	81 (55.1)
Absent (EichnerB4-C3)	66 (44.9)
Prosthetic treatment during hospitalization	
Present	53 (36.0)
Absent	94 (64.0)

FIM, Functional Independence Measure; CONUT, Controlling Nutritional Status; ESS, Eating Status Scale.

**Table 2 geriatrics-08-00104-t002:** Molar occlusal support and denture wearing status.

	Molar Occlusal Support Present (EichnerA1–B3)	Molar Occlusal Support Is Absent (EichnerB4–C3)
Denture wearer	7 (8.6%)	23 (34.8%)
Non-denture wearer	74 (91.4%)	43 (65.2%)
Total	81	66

(*p* < 0.05, χ^2^ test).

**Table 3 geriatrics-08-00104-t003:** The correlation coefficient between the FIM total score and the oral and general condition.

FIM Total Score	Age *	Sex	Stroke *	ESS *	CONUT Variant	Number of Present Teeth	Number of Functional Teeth	Molar Occlusal Support	Need of Prosthetic Treatment
correlation coefficient	0.170 *	0.011	−0.239 **	0.625 **	−0.095	−0.006	0.129	−0.013	0.059
*p*-value	0.039	0.899	0.004	<0.001	0.253	0.947	0.120	0.880	0.481

Sex: 0, male; 1, female. Stroke: 0, stroke; 1, no stroke. ESS: 0, tube feeding only; 4, oral feeding (unmodified) CONUT variant score: 0, normal; 12, severe. Molar occlusal support: 0, present; 1, absent. prosthetic treatment: 0, no prosthetic treatment needed during hospitalization; 1, prosthetic treatment is needed. *: *p* < 0.05; **: *p* < 0.01.

**Table 4 geriatrics-08-00104-t004:** Multiple regression analysis results with the FIM total score as the objective variable.

Explanatory Variable	B	SE	β	*p*-Value	95% CI	VIF
Age *	0.340	0.153	0.165	0.028	0.037 to 0.643	1.298
Sex	−1.821	3.679	−0.033	0.621	−9.095 to 5.452	1.040
Stroke	−6.913	3.668	−0.130	0.062	−14.165 to 0.338	1.120
ESS score *	9.320	1.211	0.522	<0.001	6.926 to 11.714	1.086
CONUT variant score *	8.053	2.403	0.243	0.001	3.303 to 12.803	1.235
Number of functional teeth *	0.417	0.194	0.151	0.033	0.033 to 0.801	1.170
Need of prosthetic treatment	2.207	4.019	0.040	0.584	−5.740 to 10.153	1.241

Sex: 0, male; 1, female. Stroke: 0, stroke; 1, no stroke. ESS: 0, tube feeding only; 4, oral feeding (unmodified) CONUT variant score: 0, normal; 12, severe. Need of prosthetic treatment: 0, no prosthetic treatment needed during hospitalization; 1, prosthetic treatment is needed. * *p* < 0.05.

## Data Availability

The datasets generated or analyzed during this study are available from the corresponding author upon reasonable request.

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
