# Peer review of "The Association between Oral Health Management and the Functional Independence Measure Scores at the Time of Admission of Inpatients to a Convalescent Hospital"

_geriatrics, 2023, doi:10.3390/geriatrics8050104_

Round 1

Reviewer 1 Report

This is an important article that suggests the importance of dental intervention, especially denture treatment, in convalescent rehabilitation hospital. This study is worthy of publication. Please consider revising the following points.

Please unify the terminology about convalescent rehabilitation hospital.

recovery rehabilitation hospital convalescent rehabilitation hospital or convalescent hospital

Materials and Methods

2.2. Outcomes

P2L89 convalescent ward convalescent rehabilitation ward

P2L90 OHAT

OHAT should be written without abbreviation since it is the first time it appears, and a summary should be included as well as other measures.

P3L121-122

This signified the necessity assessed by dentists and the needs of patients for prosthetic treatment.

A little confusing. Because of the initial dental evaluation, prosthetic treatment has probably not yet been initiated in the convalescent rehabilitation hospital.

Results

3.1. Participant characteristics

P3L138

Please also include the percentage of ≥ 65 years older participants.

P4L158-159 This indicated that while many patients required dentures, denture 158 treatment was requested in relatively fewer cases (Fig. 1).

Since this sentence is a discussion, I think it is appropriate to write it within the discussion section.

Figure 1.

Oral examination prior to swallow angiography Oral examination prior to videofluoroscopic examination of swallowing

Dental hygiene management Oral hygiene management

3.2. Association between level of independence and oral and general condition

P5L172-174 Table 3.

It is incorrect to examine the correlation coefficient between categorical variables such as sex, Stroke, Molar occlusal support and prosthetic treatment, and FIM total.

It would be better to include also the number of present teeth and molar occlusal support in the multiple regression analysis (if there is no problem with VIF) and present only the results of multiple regression analysis without correlation coefficient analysis.

Table 4.

There are two B”s. I think one is "β” (beta).

Discussion

P6L196-199

Please reconsider this sentence as it relates to correlation coefficient analysis.

P7L231-232 The results of multiple regression analysis clarified that the number of functional teeth is associated with daily function as well as eating normal food, and good nutritional status.

This sentence means,

“number of functional teethdaily function, eating normal food, and good nutritional status”,

but,

“number of functional teeth, eating normal food, and good nutritional statusdaily function”,

isn’t it?

P7L238-240 However, this study did not reveal a relationship between oral deterioration and malnutrition in patients admitted to the convalescent hospital, nutritional status and oral health management, or oral intake and FIM score in stroke patients in the convalescent period.

I think these analyses have not been performed except for the FIM score.

Please check and correct some English sentences that are difficult to understand.

Author Response

Dear. Reviewer

Thank you for your kind review.

We almost agreed your comments and revised according to your comments.

Please see the attachment file.

Sincerely,

Junichi Furuya and Ryuzo Hara

Reviewer 2 Report

In this manuscript entitled “Association between Oral Health Management and Functional Independence Measure scores at admission of inpatients in a convalescent hospital”, the authors propose that Oral hygiene management through appropriate medical and dental coordination supports improved daily living functions.

The major finding of the study is oral health management is important in convalescent hospitals to support independence, which has not previously been reported.

This is an important paper arguing the role of better nutritional status in improvement of the oral health.

However, I have one minor comment regarding the interpretation of their data.

It is unclear whether improvement of the oral health is directly associated with better nutritional status, as claimed by the authors.

There is no explanation as to why the patient was malnourished despite taking regular food orally.

Therefore, it is necessary to consider whether the effect of the oral problems are related to the amount of food eaten and the decreased appetite.

The authors should address this point in their Discussion.

Author Response

(The authors gave the same response as above.)

Reviewer 3 Report

This is an excellent research article and I really enjoyed reading it. It is wonderful that patients in a convalescent hospital have access to a dentist during their stay. This does not happen at countries outside of Japan. The authors should be congratulated for researching and then sharing the results from this wonderful dental service.

Lines 145-146 plus data in Table 1 – please check to see that ‘present’ and ‘functional’ teeth are in the correct order. I would have thought that more teeth would be ‘present’ (16.6) when compared to ‘functional’ teeth (20.8) but this is not the case.  

Table 1 – the heading is ‘Mean and SD/N (%)’ but this is not clear. Consider ‘Mean and SD / N (%)’

Minor typographical changes:

Line 122 – ‘support’ not ‘spport’

Line 199 – ‘been’ not ‘ben’

Author Response

(The authors gave the same response as above.)
